# Physiological magnetic field strengths help magnetotactic bacteria navigate in simulated sediments

Agnese Codutti[1,2,3†], Mohammad A Charsooghi[2†], Konrad Marx[4†], Elisa Cerdá-Doñate[2†], Omar Muñoz[4], Paul Zaslansky[2,5], Vitali Telezki[4], Tom Robinson[1,6]*, Damien Faivre[2,7]*, Stefan Klumpp[4]*

[1]Max Planck Institute of Colloids and Interfaces, Department Theory and Biosystems, Potsdam, Germany; [2]Max Planck Institute of Colloids and Interfaces, Department Biomaterials, Potsdam, Germany; [3]Physics Department, TU München, Garching, Germany; [4]University of Göttingen, Institute for the Dynamics of Complex Systems, Göttingen, Germany; [5]Charité - Universitätsmedizin Berlin, Department for Operative, Preventive and Pediatric Dentistry, Berlin, Germany; [6]Institute for Bioengineering, School of Engineering, University of Edinburgh, Edinburgh, United Kingdom; [7]Aix-Marseille Université, CEA, CNRS, BIAM, Saint Paul lez Durance, France

**\*For correspondence:**
tom.robinson@mpikg.mpg.de (TR);
damien.faivre@lu.lv (DF);
stefan.klumpp@phys.uni-goettingen.de (SK)

[†]These authors contributed equally to this work

## eLife Assessment

This study presents **valuable** experimental and numerical results on the motility of a magnetotactic bacterium living in sedimentary environments, particularly in environments of varying magnetic field strengths. The evidence supporting the claims of the authors is **compelling** and the study will be of specific relevance to biophysicists interested in bacterial motility.

**Abstract** Bacterial motility is typically studied in bulk solution, while their natural habitats often are complex environments. Here, we produced microfluidic channels that contained sediment-mimicking obstacles to study swimming of magnetotactic bacteria in a near-realistic environment. Magnetotactic bacteria are microorganisms that form chains of nanomagnets and that orient in Earth's magnetic field. The obstacles were produced based on micro-computer tomography reconstructions of bacteria-rich sediment samples. We characterized the swimming of the cells through these channels and found that swimming throughput was highest for physiological magnetic fields. This observation was confirmed by extensive computer simulations using an active Brownian particle model. The simulations indicate that swimming at strong fields is impeded by the trapping of bacteria in 'corners' that require transient swimming against the magnetic field for escape. At weak fields, the direction of swimming is almost random, making the process inefficient as well. We confirmed the trapping effect in our experiments and showed that lowering the field strength allows the bacteria to escape. We hypothesize that over the course of evolution, magnetotactic bacteria have thus evolved to produce magnetic properties that are adapted to the geomagnetic field in order to balance movement and orientation in such crowded environments.

## Introduction

The motility of microorganism and other self-propelled particles has been studied extensively over the last years, aiming at an understanding of the physical mechanisms of micron-scale self-propulsion and

of the rich collective behavior shown by such active particles (*Bechinger et al., 2016*; *Elgeti et al., 2015*; *Klumpp et al., 2019*). In addition, biomedical and environmental applications of functionalized self-propelled particles or microrobots are being developed (*Bente et al., 2018*; *Felfoul et al., 2016*; *Magdanz and Schmidt, 2014*). Often microorganisms in their natural habitats and microrobots in some of their envisioned applications move in complex environments rather than in a homogeneous space. Therefore, the interactions of self-propelled particles with walls and obstacles play a key role in their motility and have received considerable attention (*Bechinger et al., 2016*).

The complexity of an environment may arise on multiple scales: specifically, an environment may be geometrically complex, e.g., a porous medium, a space filled with obstacles or a maze (*Volpe et al., 2011*; *Jakuszeit et al., 2019*; *Bhattacharjee and Datta, 2019b*; *Khatami et al., 2016*), or the interaction with walls and obstacles may itself be complex and reflect a combination of hydrodynamic and steric interactions (*Berke et al., 2008*; *Li and Tang, 2009*; *Bianchi et al., 2017*; *Ostapenko et al., 2018*; *Codutti et al., 2022*) or even behavioral responses to the encounter with an obstacle, such as direction reversals (*Kühn et al., 2017*) and effects of population heterogeneity (*Junot et al., 2022*).

Both steric and hydrodynamic interactions with walls have been the subject of numerous theoretical and computational as well as experimental studies (*Berke et al., 2008*; *Li and Tang, 2009*; *Bianchi et al., 2017*; *Elgeti and Gompper, 2013*; *Ostapenko et al., 2018*; *Mousavi et al., 2020*; *Telezki and Klumpp, 2020*; *Codutti et al., 2022*). A useful approach to wall interactions is the trapping of microorganisms in specifically designed microfluidic channels (*Denissenko et al., 2012*) or in microfluidic chambers (*Ostapenko et al., 2018*; *Codutti et al., 2022*; *Bentley et al., 2022*), which is also of interest in itself to understand their behavior in confinement, which is again a common feature of complex natural habits, e.g., in porous media. Surprisingly, the behavior of microorganisms with different hydrodynamic mechanisms was seen to be rather similar under such confinement (*Ostapenko et al., 2018*; *Codutti et al., 2022*). The motility of microorganism through arrays of obstacles or through porous materials has also been addressed in several experimental and theoretical studies, but mostly in geometric arrays of pillars of the same morphologies (*Bhattacharjee and Datta, 2019a*; *Dehkharghani et al., 2023*; *Irani et al., 2022*; *Jakuszeit et al., 2019*; *Bhattacharjee and Datta, 2019b*; *Alirezaeizanjani et al., 2020*).

In addition to obstacles, the complexity of an environment may reflect directional cues such as chemical gradients or external fields, to which the microorganisms show a behavioral response such as chemotaxis, phototaxis or magnetotaxis (*Sourjik and Wingreen, 2012*; *Klumpp et al., 2019*). Yet another common factor of complexity is fluid flow for active particles that swim (*Kessler, 1985*; *Rusconi et al., 2014*; *Mathijssen et al., 2019*). Thus, understanding the motility of microorganisms in such environments requires an understanding of the interplay of these different influences, which in general could either have synergistic effects or compete with each other in determining the direction of motion of a microorganism.

Here, we study the navigation of magnetotactic bacteria through a disordered obstacle array as an example of motility in a complex environment. Specifically, we are interested in the interplay of directed motility due to the magnetic field and the randomization of motion required to navigate the obstacles.

Magnetotactic bacteria are a group of bacteria that form a chain of specific organelles called magnetosomes, which contain magnetic nanoparticles and equip the cells with a magnetic moment. As a consequence, the cells align with magnetic fields, resulting in directional swimming of the cells, powered by their flagella, along the field lines of the magnetic field. In their natural habitats, these bacteria follow the magnetic field of the Earth, so they align to fields of the order of 50 µT. The swimming of magnetotactic bacteria has been studied in free space, in the presence of oxygen gradients, in hydrodynamic flow, and in confinement to small chambers (*Bennet et al., 2014*; *Lefèvre et al., 2014*; *Mao et al., 2014b*; *Mao et al., 2014a*; *Reufer et al., 2014*; *Popp et al., 2014*; *Rismani Yazi et al., 2018*; *Codutti et al., 2022*).

Here, we study their swimming in obstacle arrays that mimic their natural environment: magnetotactic bacteria are predominantly found in micro-oxic aquatic environments, near the oxic-anoxic transition zone (*Bazylinski and Frankel, 2004*). Often, these micro-oxic regions are found in the sediment rather than in open water, so that magnetotactic bacteria typically live in a sedimentary environment that is sand or mud. We thus prepare microfluiudic environments that mimic these sediments and consist of arrays of irregularly shaped obstacles. We study the swimming of magnetotactic bacteria

through these environments and ask how a magnetic field influences their throughput, as the field directs and constrains their motion, while navigation in the obstacle array requires the exploration of other swimming directions. Based on a combination of swimming experiments in the microfluidic environments and extensive computer simulations of an active Brownian particle model (*Codutti et al., 2019*), we show that intermediate magnetic field strength of the intensity of the geomagnetic field enhance throughput, while strong fields suppress it. We show that the suppression is due to trapping of bacteria in corners of the obstacle array from which the bacteria can only escape by transiently swimming against the magnetic field direction. This situation is reminiscent of escape problems from statistical physics, but is not driven by thermal fluctuations, but rather by the interplay of active motility and interactions with the obstacle surfaces.

## Results and discussion
### Characterization of sediment and construction of sediment-mimicking microfluidic channels

To study the swimming of magnetotactic bacteria in a near-realistic sediment environment that resembles their natural habitat, we produced microfluidic channels that contained sediment-mimicking obstacles (*Figure 1*, *Figure 1—figure supplement 1*). To that end, we first characterized a sediment sample in water from the magnetotactic bacteria-rich lake Grosser Zernsee (Potsdam, Germany). We determined a 3D reconstruction of the sample using micro-computer tomography (μCT) with a resolution of 1.56 μm (*Figure 2a–d*) and analyzed the sizes of sediment particles (sand grains) and of water gaps. In total, the sample consists of 61% sand and 39% water. The distribution of particle sizes is approximately a log-normal distribution with a mean particle diameter of 46 μm (*Figure 2e*), which classifies this sediment as silt (*Boggs, 2011*). The water gap sizes follow a normal distribution with a mean of 43 μm (*Figure 2f*).

We next analyzed two-dimensional slices of the μCT images at various depths of the sample to study if any difference is observed. This analysis returns similar values of the sand-to-water percentage (62–66% from top to bottom, see *Figure 2g*, with the small difference possibly due to sedimentation of the sediment under gravity). The mean radii of circular fits to the grains varies from 68 to 62 μm from bottom to top (*Figure 2g*), further potentially indicating a slight effect of sedimentation by gravity. This analysis provides the basis for the construction of our 2D microfluidic channels.

To realistically represent the porous environment where magnetotactic bacteria live, we constructed quasi-twodimensional microfluidic channels based on the 2D μCT images at different depths (*Figure 1*). These images (shown in *Figure 1—figure supplement 2*) were used as masks (see Methods), such that each sediment grain becomes an irregularly shaped pillar in our quasi-2D structure. Bacteria thus swim through an array of pillars that act as obstacles for their motion. In addition, we generated channels with cylindrical pillars by fitting circles to the two-dimensional grain images (*Figure 1*, *Figure 1—figure supplement 2*). These rounded-shaped pillars of different diameters are used for a comparison of our experiments with computer simulations below.

### Swimming of magnetotactic bacteria through sediment-mimicking obstacle channels

With the sediment-mimicking channels at hand, we studied the motion of magnetotactic bacteria through those channels. A droplet of culture medium containing bacteria was injected into the entrance channel, which is placed under a custom-designed microscope (*Bennet et al., 2014*), with a controlled magnetic field of either 0 μT (field canceled), 50 μT (intensity of the geomagnetic field), or 500 (one order of magnitude larger than the geomagnetic field) μT (*Figure 3a*, *Figure 3—figure supplement 1*). In all cases, no external flow was imposed.

We then imaged the entrance and exit regions of the microfluidic channel in an alternating fashion (*Figure 3a*), with snapshots every 20 s and counted the number of bacteria in these two regions. These counts are noisy and some bacteria pass the channel rapidly, within the first few snapshots. The delay time after which the first bacteria reach the exit region of the channel could, therefore, not be reliably quantified. Instead, we determined the cumulative density of bacteria in the exit and in the entrance regions of the channel (*Figure 3b*) and used their ratio, determined 30 min after injecting the bacteria, as a measure of the throughput of bacteria through the channel (see Methods). We

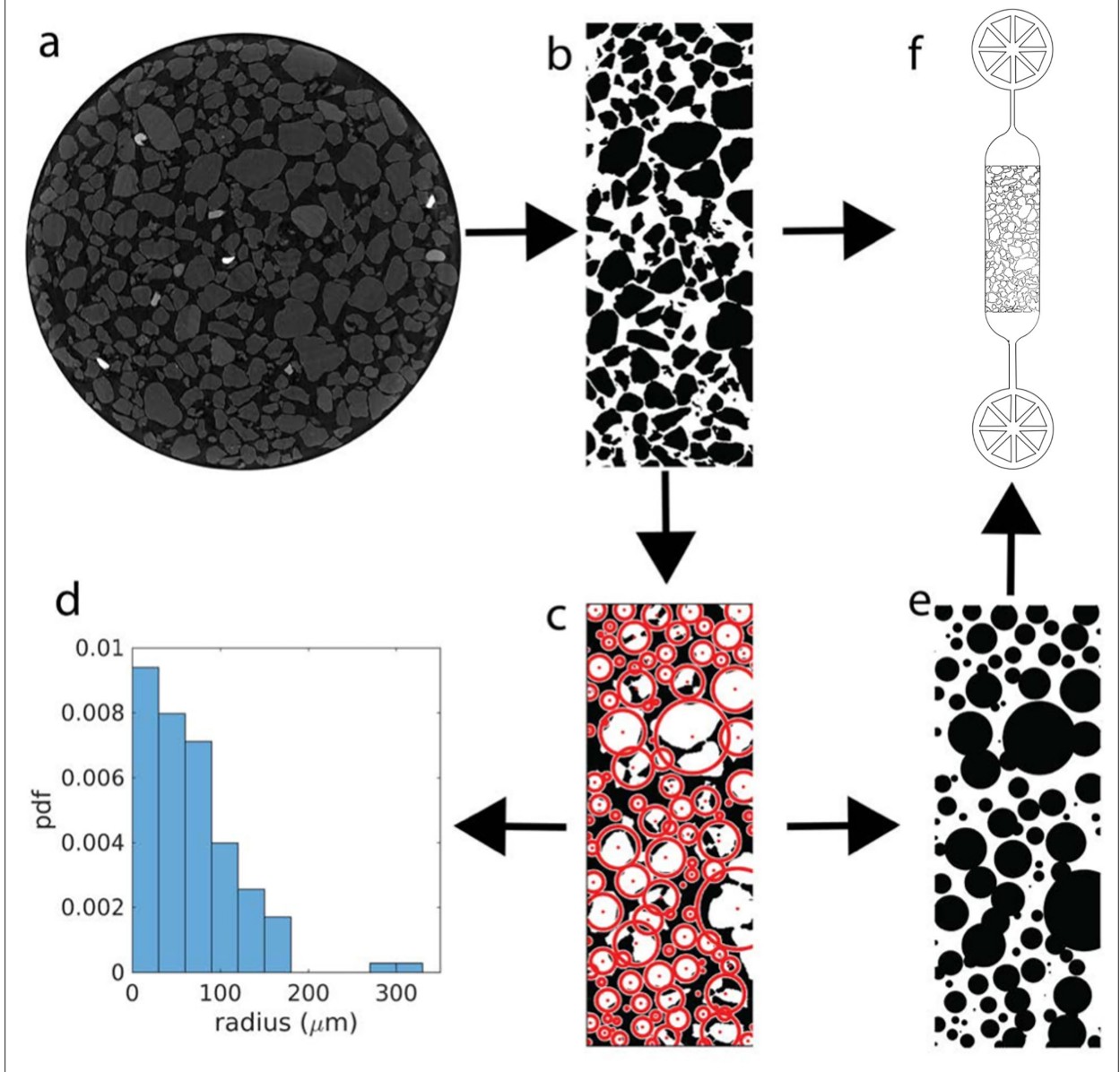

**Figure 1.** Construction of sediment-mimicking obstacle channels. (**a**) Micro-computer tomography slices of sediment sampled (diameter 4mm) are cropped out to ignore border effect due to the reconstruction and binarized (**b**). The binarized images are then fitted with circles (**c**) for statistical analysis (grain size distribution in d) and for construction of arrays of pillar-shaped obstacles (**e**). The images of irregular grains (**b**) and fitted circles (**e**) are used as masks to design microfluidic channels with obstacle arrays (**f**), in which the bacteria are injected in the middle channel (3.5 mm × 1.2 mm × 10 μm) where they encounter the obstacles.

The online version of this article includes the following figure supplement(s) for figure 1:

**Figure supplement 1.** Two microfluidic channels with irregularly shaped (**A, C**) obstacles and the corresponding channels with rounded obstacles (**B, D**).

**Figure supplement 2.** Masks for all obstacle arrays used in this study.

determined the throughput in this way for 20 different channels, 10 with irregularly shaped obstacles and 10 with cylindrical pillars (*Figure 1—figure supplement 2*). We obtained very similar results for channels with cylindrical pillars and with irregularly shaped pillars (*Figure 3—figure supplement 2*). However, we found the strength of the magnetic field to have a strong impact on the throughput, with a non-monotonic dependence on the field strength (*Figure 3c*). The maximal throughput was obtained for the intermediate field strength of 50 μT, while without a field as well as with a field of 500

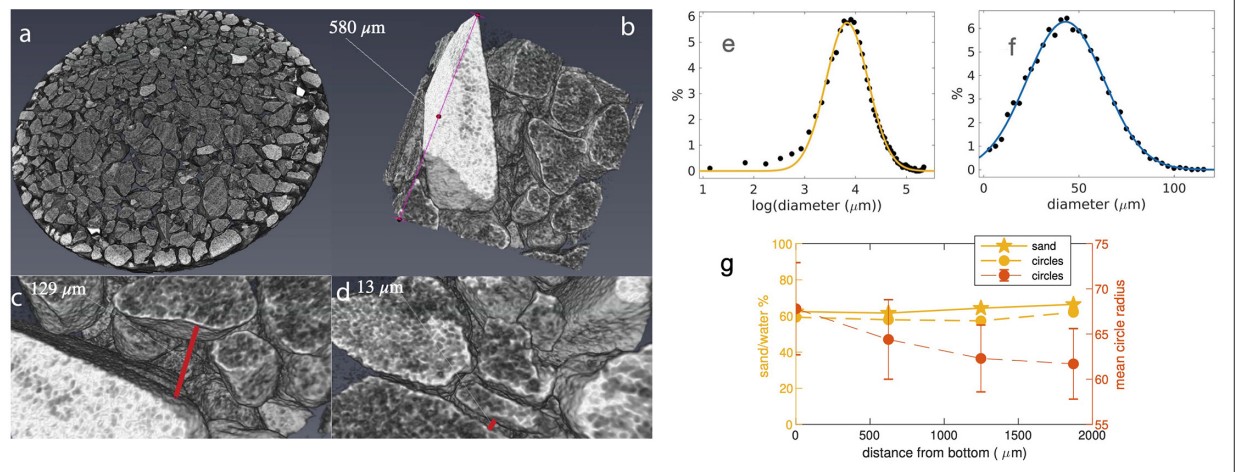

**Figure 2.** Analysis of micro-computer tomography images of a sediment sample in water. (**a**) Slice of the sediment sample, (**b**) cube from the center of the cuvette, (**c**) and (**d**) examples of water gaps rendered with the Amira software. (**e**) Distribution of the smallest grain dimension (i.e. trabecular thickness analyzed with CTAn) with Gaussian fit $a \exp(-(x - \mu)^2/c^2)$, $\mu = 3.83$, $c = 0.5903$, corresponding to a mean sand grain diameter of 46 µm. (**f**) Distribution of the water gap size (i.e. trabecular separation) with Gaussian fit $a \exp(-(x - \mu)^2/c^2)$, $\mu = 42.94$ µm, $c = 28.66$ µm. (**g**) Percentage of sand over water as function of the distance from the bottom of the cuvette as obtained from 2D slices of the µCT images (yellow filled line) and for the equivalent fit with circles (yellow dashed line). The red points show the mean radius of the fitted circles (± standard error of the mean) in µm (right axis).

µT the throughput was smaller. By contrast, for a channel without obstacles, the throughput increased monotonically with the field strength (*Figure 3—figure supplement 3*). This observation suggests that a magnetic field of a similar strength of the Earth's field enhances the flux of bacteria through the obstacle channel, but a too strong or null magnetic field inhibits it.

## Simulations of bacterial swimming through obstacle channels

To gain further insight into how magnetotactic bacteria navigate through these complex environments, we compared our experimental findings to numerical simulations. To this end, we modeled the magnetotactic bacteria as dipolar active Brownian particles that align in an external magnetic field and interact with obstacles and walls present in the simulation box (see Methods).

Most parameters of the model have been directly measured, only the interactions with the obstacles have to be parameterized by matching simulations to experimental data. When interacting with an obstacle or a wall, the active particles are subject to a repulsive force as well as a reorienting torque (*Telezki and Klumpp, 2020*). The latter is parameterized by an interaction parameter $\alpha$, the only crucial free parameter of our model. $\alpha$ has the units of length and is expected to be a fraction of the size of a bacterium. To adjust this parameter, we observed the motion of magnetotactic bacteria near the obstacles. When a bacterium meets an obstacle, it slides along the contour of that obstacle for some distance before leaving it again, as shown in *Figure 4a*. Similar behavior has been observed in other microorganisms (*Rothschild, 1963*; *Berg and Turner, 1990*; *Tuson and Weibel, 2013*). We observe the same behavior in the numerical simulations (*Figure 4b*). Here, the sliding distance along an obstacle is greatly influenced by the surface torque parameter $\alpha$ (*Figure 4—figure supplement 1*). Therefore, we calculated distributions of the sliding distances for different values of $\alpha$ and compared these distributions to the one obtained from the experimental observations (see *Figure 4c*). A good match is obtained for $\alpha = 0.2\,\mu m$. Therefore, all the following numerical simulations were carried out with the interaction parameter set to $\alpha = 0.2\,\mu m$.

When analyzing sliding distances from obstacles of different sizes and thus different curvatures, we saw only a weak dependence on the curvature in both experimental and simulated trajectories (*Figure 4—figure supplements 1 and 2*). However, our obstacles are all much larger than the size of the bacteria, so we cannot exclude a curvature dependence for obstacles comparable to their size. We noticed that in the experiments, the bacteria appeared to be slower while sliding along the walls compared to free swimming. This effect was neglected in the simulations.

Next, we simulated the motion of active dipolar particles representing the bacteria through channels with arrays of spherical obstacles (which may overlap as in the experiments with cylindrical pillars).

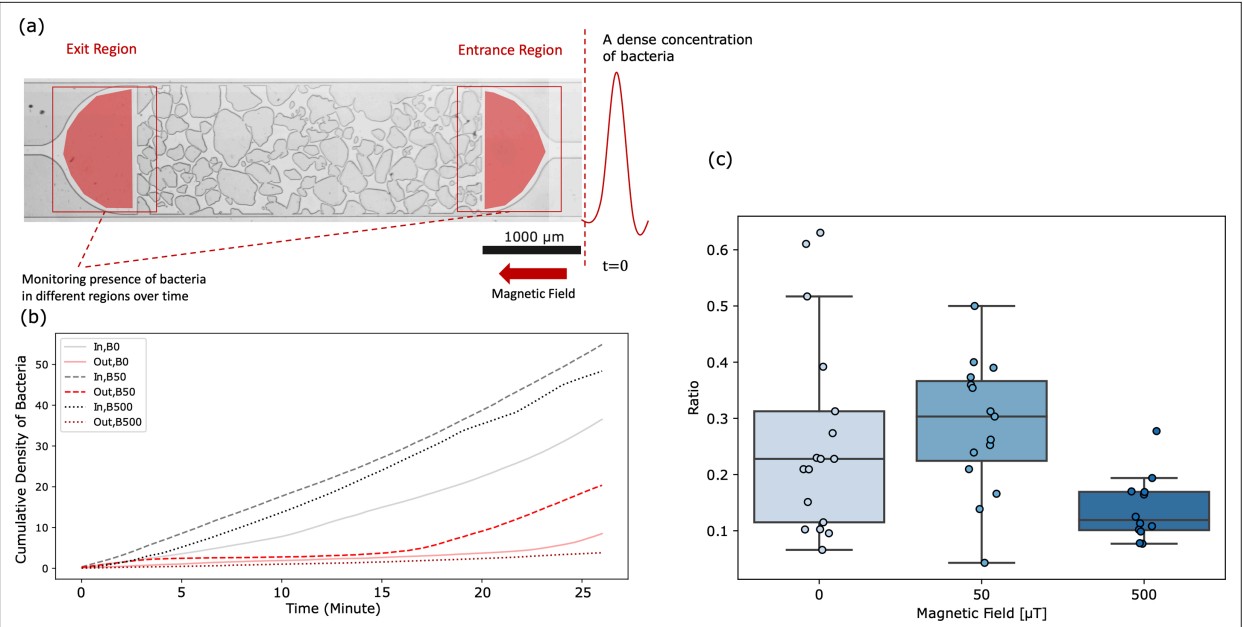

**Figure 3.** Swimming of magnetotactic bacteria through obstacle channels. (**A**) View of a channel, in which the entrance (IN) and exit regions (OUT) are indicated, in which the bacterial density was monitored. Bacteria enter the IN region through an inlet that connects to a syringe containing bacteria. The magnetic field points to the left. (**B**) Cumulative bacterial intensity (measuring the cumulative arrival of bacteria) in these two regions, for three different field strengths. (**C**) Bacterial throughput, quantified by the ratio of the cumulative intensities in the OUT and IN regions as function of the magnetic field. The reduction of throughput by $B$ = 500 µT compared to $B$ = 50 µT is significant (Mann-Whitney test, $p < 10^{-3}$), the increase between $B$ = 0 and $B$ = 50 µT is weakly significant ($p = 0.028$), but agrees with the observation for a channel without obstacles in which the throughput increases monotonically with increasing field strength (*Figure 3—figure supplement 3*).

The online version of this article includes the following figure supplement(s) for figure 3:

**Figure supplement 1.** Sketch of the setup showing the microfluidic system (blue), the inlet connecting to a syringe with bacteria and growth medium, one of the three pairs of Helmholtz coils to control the magnetic field, and a detailed view of the microfluidic system.

**Figure supplement 2.** Comparison of bacterial throughput as a function of the magnetic field (0, 50, 500 µT) in channels with irregular (sand-like) obstacles and in channels with rounded (pillar-like) obstacles.

**Figure supplement 3.** Bacterial throughput in a channel without obstacles.

We varied the magnetic field strength and, to obtain a detailed picture of the swimming paths along which bacteria navigate through these channels, simulated channels with differently arranged obstacles, using the same obstacle arrays as in the experiments. In each simulation, we placed 1000 particles at the entrance end of the channel and calculated their individual trajectories for $N_{steps} = 1 \times 10^6$ steps (corresponding to approximately 30 min).

*Figure 5* shows the time-averaged local densities of bacteria during the simulation in one such channel with an external magnetic field of 50 µT pointing from left to right. In this heat map, shades of blue show the density of bacteria that did not arrive at the exit (right end of the channel) within the simulation time, while shades of red show the density of bacteria that did arrive, with purple resulting from a superposition of both color maps. Overall, bacteria are seen to move in a directed fashion, in the direction of the magnetic field (left to right). When they encounter a (single) obstacle, they move along its surface and then leave it, typically tangentially to the surface and in the direction of the magnetic field. In this way, the channel's geometry organizes the trajectories into 'natural paths.' However, when bacteria encounter overlapping obstacles, they can get trapped. Some examples of such traps are marked by green boxes in *Figure 5*. To escape a trap, bacteria need to reorient temporarily against the direction of the magnetic field. While some traps allow a fraction of the bacteria to escape and eventually arrive at the exit (purple trajectories that turn red), others do not (blue trajectories). Blue trajectories typically end in a trap of the latter type.

*Figure 6a* shows corresponding trajectories (again plotted as heat maps) for different strengths of the magnetic field. Increasing the magnetic field strength has two opposing effects: On the one hand,

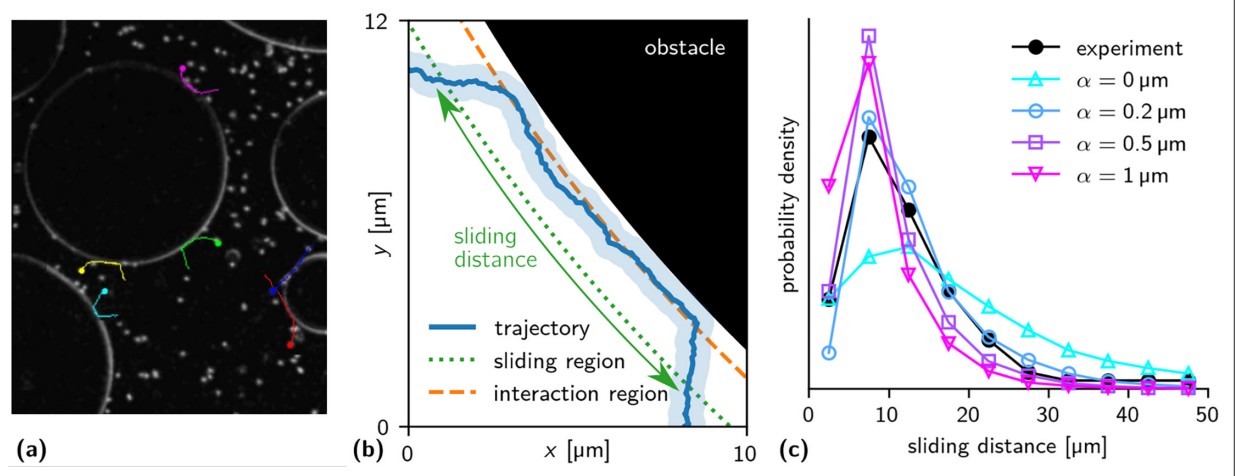

**Figure 4.** Sliding of bacteria on obstacle surfaces: (**a**) Experimental trajectories of sliding bacteria ($B = 0$). (**b**) Trajectory of sliding particle in the simulation. The light blue region shows the spatial extent of the particle. Sliding is defined as motion in the sliding region (up to 2 μm from the surface, dotted line), provided that the particle also reaches the region where interactions with the obstacle take place (up to ≈ 0.6 μm from the surface, dashed line). The sliding distance is the distance covered tangentially to the surface. (**c**) Histogram of sliding distances as measured in the experiment (filled black circles) and the simulation (empty markers, for different values of the wall torque parameter $\alpha$). We find good agreement for $\alpha = 0.2$ μm.

The online version of this article includes the following figure supplement(s) for figure 4:

**Figure supplement 1.** Sliding distance as a function of the pillar radius as obtained from simulations for different wall torque parameters $\alpha$.

**Figure supplement 2.** Dependence of the sliding distance on the curvature of the pillar as obtained from the tracking bacteria in obstacle channels with cylindrical obstacles.

bacteria are on average more aligned with the direction of the magnetic field and the observed paths through the channel become more focused as their persistence length increases with increasing field. As a consequence, the bacteria are expected to cross the channel faster. For very weak fields, the bacteria explore the channel in an effectively diffusive manner; for the case of 50 μT, we observed one dominant path through most of the channel. On the other hand, bacteria become more susceptible

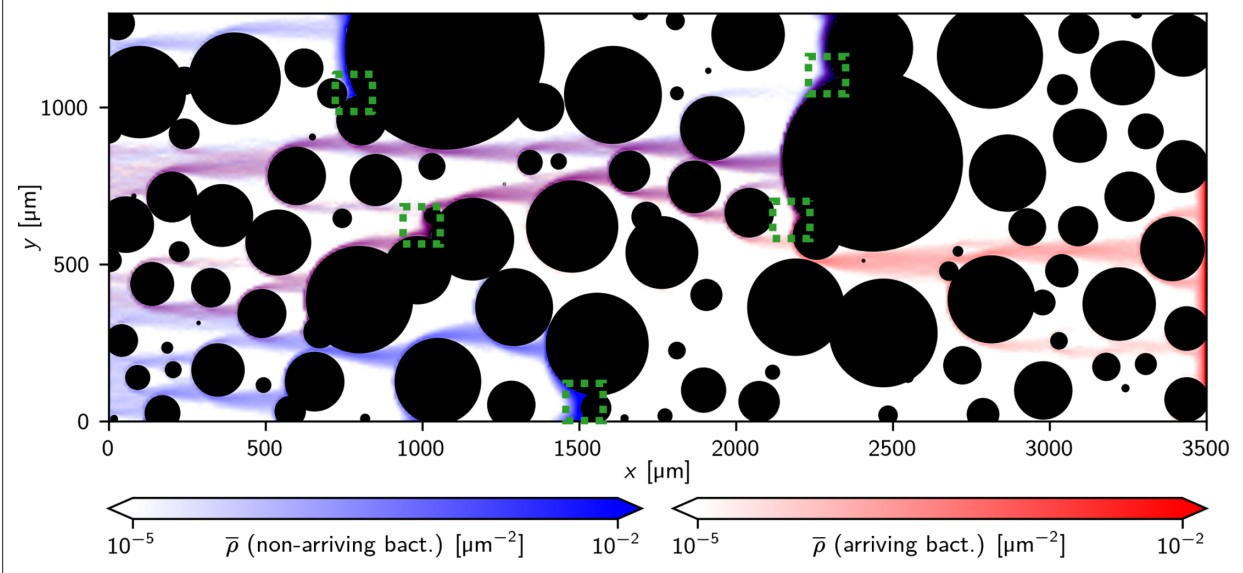

**Figure 5.** Simulated motion of bacteria in an obstacle channel: Heat maps of the time-averaged density of bacteria from simulations of 2000 bacteria for 30 min with a magnetic field $B = 50\,\mu\text{T}$, pointing from left to right. The blue map represents bacteria that did not arrive at the right end of the simulation box during the simulated time the red heat map represents those that did. The interplay between magnetic field and the channel's geometry creates natural paths through the channel. These paths can include traps or end in traps that are formed by overlapping obstacles (examples are indicated by the green boxes).

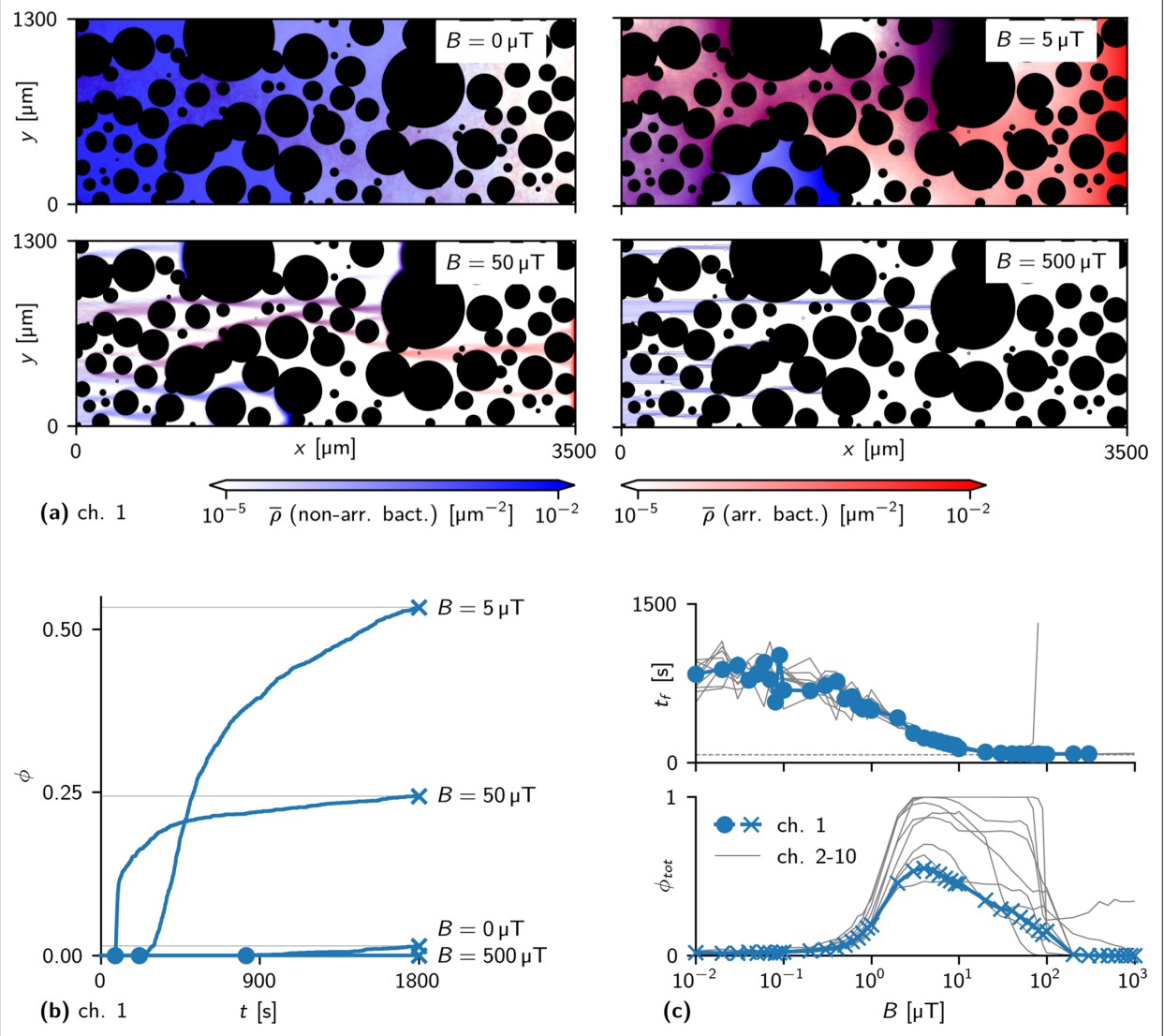

**Figure 6.** Effect of magnetic field strength on simulated swimming through obstacle channels. (**a**) Motion of bacteria in a channel for different values of the field strength $B$ (heat maps of the densities of arriving and non-arriving bacteria, blue and red, respectively, as in *Figure 5*) (**b**) Fraction $\phi$ of simulated bacteria that have arrived up to time $t$ for different $B$. Circles mark the first-arrival times $t_f$; crosses with thin lines mark total arrival fractions $\phi_{tot}$. The curve for $B = 500$ μT remains at 0. (**c**) First-arrival times $t_f$ (top) and total arrival fraction $\phi_{tot}$ (bottom) as a function of the field strength $B$. With stronger fields, $t_f$ converges to the first-arrival time of a persistent swimmer in an empty channel (dashed gray line). $\phi_{tot}$ shows a peak at an intermediate (optimal) field strength, which can be explained by the interplay of two opposing effects that arise with stronger fields: effectively faster motion in the direction of the field and higher susceptibility to trapping. Blue data points show the results for the channel shown in (**a**), gray lines show results in other channels.

to trapping with increasing field. For a field of 500 μT, no simulated bacterium crossed the channel within the simulation time and the path that was dominant at 50 μT now ends in a trap, from which the bacteria can no longer escape at 500 μT. This negative effect of the magnetic field arises because escape from traps requires that the bacteria transiently orient against the field. These effects thus have opposing influences on the throughput of the channel and provide an explanation for our observation that intermediate field strengths generate the highest throughput.

To quantify these observations, we determined the cumulative arrival of bacteria in the exit area, $\phi_{tot}$, which is plotted in *Figure 6b* as a function of time. This quantity also exhibits the non-monotonic behavior as a function of the magnetic field. We then performed extensive numerical simulations for a wide range of field strengths and for 10 different channels with different obstacle arrangements. We quantified the time of the first arrival (indicated by circles in *Figure 6b*) and the overall fraction of

bacteria that arrived until the end of the simulation time $\phi_{tot}$ (crosses in **Figure 6b**) as functions of the field strength (**Figure 6c**). The first arrival time decreases with increasing field strength, however, the poor throughput for weak fields is mostly due to the slow arrival of the bulk of the bacteria, while some bacteria still cross the channel on an almost direct path. The fraction of arrivals, by contrast, shows the non-monotonic behavior indicated by our experimental results. Results for different channels (gray lines in **Figure 6c**) are qualitatively similar. Quantitatively, the arrival fraction $\phi_{tot}$ shows considerable dependence on the channel geometry, with maximal arrival fractions between 50 and 100% and maximal throughput in the range between a few and a few tens of μT. By contrast, the first-arrival time $t_f$ seems not to be affected by channel geometry (**Figure 6c**). The observation of a pronounced maximum in the throughput of the channel as well as of the variability in throughput between channels in our simulations agree well with the experimental observations. However, the simulations show lower throughput as well as lower variability in throughput than the experiments in the case without a magnetic field. We attribute this difference to different initial conditions in experiment and simulations. In the simulations, all bacteria start simultaneously at the beginning of the obstacle channel, while in the experiments they enter from the inlet and continue to do so during the experiments. The difference is expected to be most pronounced in the case of non-directional motion, i.e., for $B = 0$.

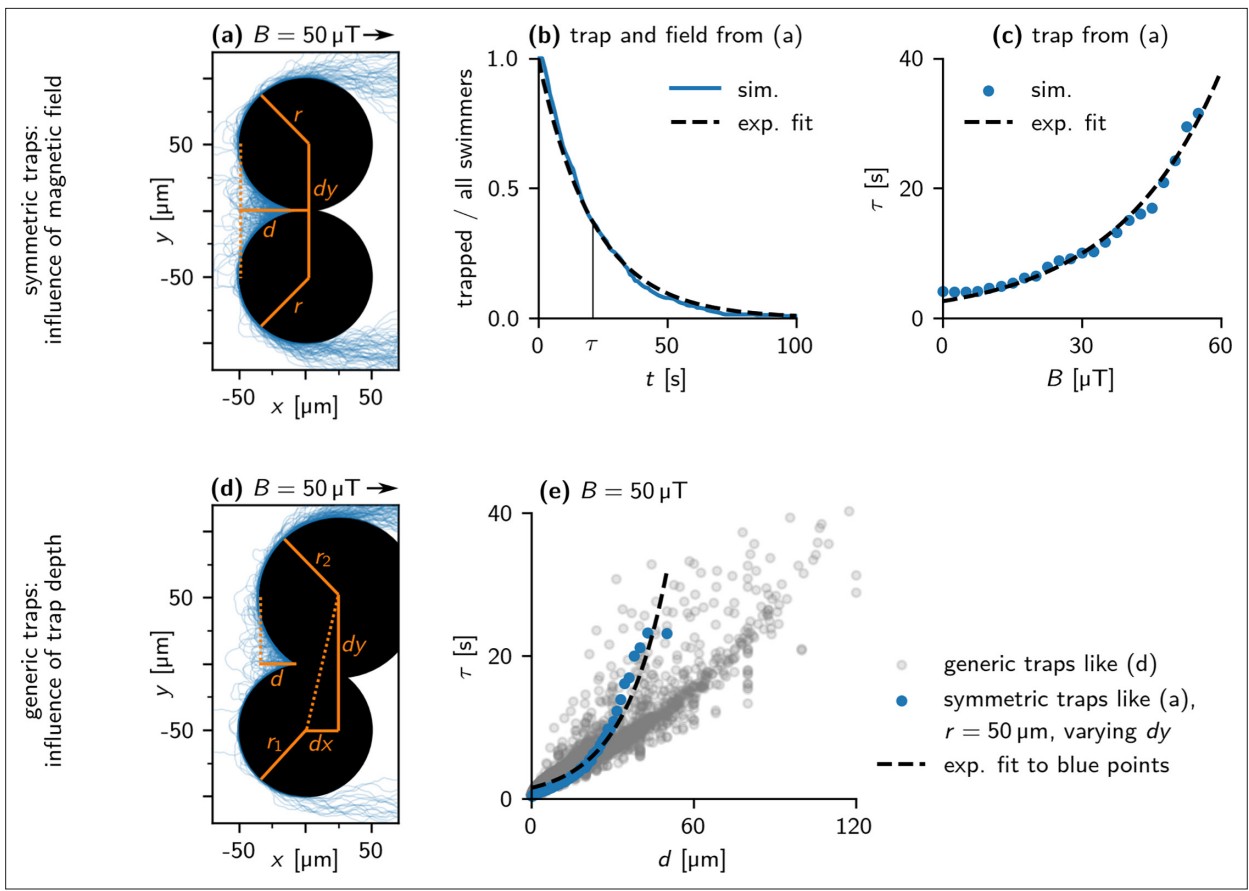

**Figure 7.** Escape from traps - Simulation. (**a**) Trajectories of 100 bacteria escaping from the apex of a symmetric trap ($B = 50$ μT). The trap consists of two obstacles of radius 50 μm, in a distance of $\Delta y = 100$ μm. This results in a trap depth $d = 50$ μm. (**b**) Distribution of escape times of bacteria from the trap in (**a**). An exponential curve $\exp(-t/\tau)$ (dashed line) is fitted to the data, with mean escape time $\tau = 22$ s. (**c**) Dependence of the mean escape time $\tau$ on the field strength $B$ for the trap in (**a**). An exponential curve, $\tau_{0;1} \exp(B/\overline{B})$, is fitted to the data (dashed line). (**d**) Trajectories of 100 bacteria escaping from the apex of a generic trap ($B = 50$ μT). (**e**) Dependence of $\tau$ on $d$ for $B = 50$ μT. Blue points show symmetric traps with both obstacles of radius 50 μm, similar to the trap in (**a**), gray points show a representative sample of all possible traps. An exponential curve, $\tau_{0;2} \exp(d/\overline{d})$, is fitted to the blue data points (dashed line).

## Escape from traps

Our simulations indicate that trapping of bacteria in the corners between overlapping obstacle pillars plays a crucial role for the non-monotonic dependence of the bacterial throughput on the magnetic field. We, therefore, investigated the escape from traps in more detail by simulating a large number of trap configurations with different geometric features. We first considered symmetric traps as shown in *Figure 7a* and varied the strengths of the magnetic field and the trap depth *d* (by varying the overlap between two circular pillars with fixed radius *r*). In this case, particles escape symmetrically in both directions, with an escape time that is, in a good approximation, exponentially distributed (*Figure 7b*). Varying the magnetic field strength results in an exponential dependence of the escape time on the field strength (*Figure 7c*). This result indicates that the escape from a trap is similar to the classical problem of an escape from a potential well due to thermal fluctuations (*Kramers, 1940*; *Hänggi et al., 1990*), even though the underlying fluctuations that orient the particle against the field are not of thermal origin here, but mostly result from the interaction of the active particle with the obstacle wall. Indeed, thermal-like distributions have been found for a related problem, the sedimentation of active particles (*Tailleur and Cates, 2009*; *Palacci et al., 2010*), specifically an exponential density profile characterized by an effective temperature. Such a description is, however, not generally possible (*Tailleur and Cates, 2009*). Likewise, we see an approximately exponential dependence on the trap depth *d* (blue data points in *Figure 7e*). We also generated a library of traps without the symmetry restriction by choosing the pillar radii and the distance of their centers (parallel and perpendicular to the field) randomly. An example with corresponding escape trajectories is shown in *Figure 7d*. This set of traps also results in a roughly exponential increase of the escape time $\tau$ with

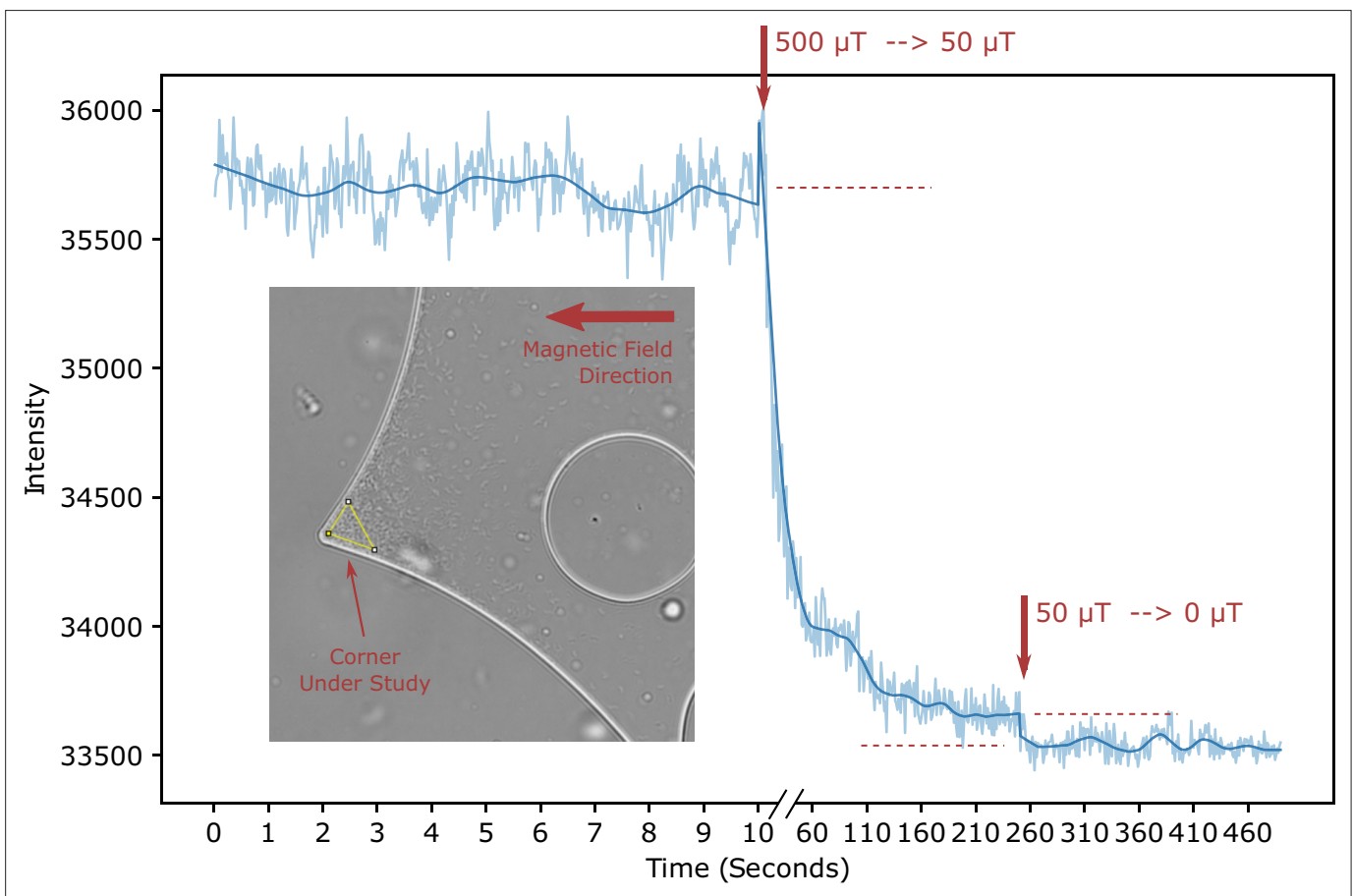

**Figure 8.** Escape from trap - Experiment. The bacterial density in a trap (indicated by the yellow triangular area in the inset) is monitored over time. At the time points indicated by the arrows, the field strength is reduced (from 500 µT to 50 µT and from 50 µT to 0 µT, respectively), allowing the escape of trapped bacteria.

trap depth, however, with large variability between different traps of the same depth (gray data points in *Figure 7e*).

Finally, we asked whether trapping of bacteria in corners is directly visible in our experiments. For that, we focused on one trap, indicated by the triangular area in *Figure 8* (inset), and tracked the bacterial density in that area over time. Indeed, we could see bacteria trapped in this corner as indicated by a constant bacterial density in the trap. We then reduced the field strength in two steps, first from 500 µT to 50 µT and then to 0 µT. After both steps, we could observe a reduction of the bacterial density (*Figure 8*), indicating the expected release of bacteria from the trap upon reduction of the field and providing direct evidence for trapping in our experiments.

## Concluding remarks

In this study, we have used experiments in microfluidic channels and particle-based simulations to address the swimming of magnetotactic bacteria through an obstacle channel mimicking their natural habitat, sediment of a lake. We have shown that the efficiency with which the bacteria navigate through the obstacle array depends on the strength of the magnetic field. For weak fields, the bacteria motion is effectively diffusive and throughput is low, as the bacteria explore the channel in a largely random motion. For high fields, bacteria do not cross the channel as they get trapped in the corners that are created by overlapping cylindrical pillars. To escape from traps, they need to swim against the field, which becomes prohibitively difficult for strong fields. At intermediate field strength, notably comparable to the magnetic field of the Earth that they encounter in the natural habitat, the bacterial throughput is maximal. Hopping between traps has been described as a generic feature of active motion in porous environments (*Kurzthaler et al., 2021*). In our case, however, the escape from the traps does not only depend on the intrinsic properties of the bacteria's swimming, but also the strength of the magnetic field and their interaction with the obstacle walls.

Our observation suggests that the ability to swim against the magnetic field, which usually directs the motion of the magnetotactic bacteria may be essential under some conditions such as the escape from traps studied here. This ability requires that the interaction with the magnetic field is not too dominant. Since in the natural habitat, the strength of the field is given by the magnetic field of the Earth and thus fixed, this might constrain the magnetic moment of magnetotactic bacteria. As a consequence, their alignment with a magnetic field of the strength of the Earth's field will not be perfect, thus reducing the efficiency of magnetic steering. Indeed, the alignment of different strains of magnetotactic bacteria with a field comparable to the Earth field has been observed to be considerably below 100% (*Klumpp et al., 2019*).

The ability to swim against the direction of a magnetic field is also important for magneto-areotactic band formation (*Codutti et al., 2019*). Contrary to the situation here, band formation requires swimming against the field direction for an extended amount of time. This is achieved by directional reversals, i.e., backward swimming that allows the bacteria to swim against the magnetic field without antiparallel alignment of their magnetic moment and the field. For escape from a trap, only transient swimming against the field is needed, so direction reversals may be less important here, even though they can also provide a mechanism for escape. In our previous work, we have shown that reversals are rare in *M. gryphiswaldense* under confinement and that they are not induced by collisions with walls (*Codutti et al., 2022*). This may, however, be different in different species and could for example depend on the mode of propulsion (*Kühn et al., 2017*), which is rather diverse in magnetotactic bacteria (*Lefèvre et al., 2014*). In general, it will be interesting to see how the competition between magnetic directionality and trapping is resolved in different species. Our microfluidic obstacle channel and our simulations provide tools for such future studies.

## Methods

**Key resources table**

| Reagent type (species) or resource | Designation | Source or reference | Identifiers | Additional information |
|---|---|---|---|---|
| Strain, strain background *Magnetospirillum gryphiswaldense* | MSR-1 | *Schleifer et al., 1991* | MSR-1 | Obtained from Schüler Lab (Univ. Bayreuth, Germany) |
| Software, algorithm | Simulation code | This paper | | See Methods, section Model and simulations |
| Other | Microfluidic channels | This paper | | See Methods, section Microfluidic channel |

## Characterization of sediment

### Sample harvesting and preparation

The sediment sample was collected in the Großer Zernsee lake (Potsdam, Germany), from the first layer of sediment (5 cm) in the shallow water nearby the shore. Macroscopic organic matter was manually removed. The sand was stored in lake water, and shaken before sample preparation to avoid strong sedimentation effects. The sample of sand in water was placed in a plastic cuvette of 4 mm in diameter, with a layer of water covering the top of the sand.

### Micro-computer tomography

The micro-computer tomography scan was performed with the SkyScan 1172 scanner. For the scans, the following setting were used: X-ray source 89 kV, 112 µA, Image Pixel Size (µm)=1.56, Exposure (ms)=1400, Rotation Step (°)=0.150, with frame averaging. Raw data were reconstructed using NRcon software (Version 1.6.10.4). For the reconstruction: Pixel Size (µm)=1.56202, Reconstruction Angular Range (deg)=360.00, Angular Step (deg)=0.1500, Ring Artifact Correction = 10, Smoothing = 0, Filter cutoff relative to Nyquist frequency = 100, Filter type description = Hamming (Alpha = 0.54), Beam Hardening Correction (%)=70.

### 3D statistical analysis with CTAn

For the statistics, the first 1316 bottom slices were used (2 mm of sample), to avoid slices not completely filled by sand and distortion effects that were seen in the top slices. The microCT images were processed by thresholding with Otsu method, despeckled (for white and black speckles, to reduce the noise) and a Median filter was applied with the the CTAn software (Version 1.16, Brucker). The 3D statistics regarding the grain-size distribution (trabecular thickness) and the water-gap distribution (trabecular separation) was obtained by the CTAn software (Version 1.16, Brucker). In this process, the grain sizes are quantified by the smallest dimension of the sediment grains. The distributions were fitted with a MATLAB fitting tool, using a Gaussian fitting model $a\exp(-(x-\mu)^2/c^2)$. For the grain-size, the fitting was performed on the distribution of the logarithm of the size, while for the water-gaps the fitting was performed on the gap size. The 3D visualization was done with the Amira software.

### 2D statistical analysis

#### 2D image preparation

The 2D slices obtained from the microCT reconstruction at different regular depths (0, 624, 1248 and 1872 µm) from the bottom of the cuvette, *Figure 1a* were first processed with ImageJ: the images were cropped in the center to avoid the border effects, they were binarized to get black and white masks and a Median filtering was applied for smoothing (*Figure 1b*).

#### Circle fitting

Subsequently, the images were analyzed by a custom-made MATLAB program (*Figure 1c*): erosion was applied to separate the grains from each other (settings: diamond shape and radius 9 pixels). Subsequently, the centroid function provided the center of each grain and the radius of a circle with equivalent area.

#### Grain size statistics

The radii of the circular fit were used for calculating the statistics of the slice (*Figure 1d*). The percentage of sand was calculated as black pixels number over total number of pixel times 100. While

the 3D analysis method estimates the smallest dimension of the grain and thus underestimated grain size, the 2D method slightly overestimates the biggest dimension of the grain (see *Figure 2g*).

## Microfluidic channel
### Mask template preparation
The 2D microCT binarized and processed images (*Figure 1b*), and the binarized images (*Figure 1e*) obtained from the circle fitting (*Figure 1c*) were adapted to avoid air bubble formation and to improve water perfusion, with water-gaps manually increased in critical points. These binary figures were used as a template to design the photomask in AutoCAD 2015 obtaining 1330 µm-wide microchannels *Figure 1f*. The design of our microfluidic systems also included channels parallel to the obstacle channel on both sides that could be used to impose chemical gradients, these were, however, not used in the present study.

### Master mould production
A high-resolution chrome photomask was obtained from Compugraphics Jena GmbH and was used to produce a master mould with $10\,\mu m$ deep features. The master mould was fabricated by baking a silicon wafer at 200°C for 20 min and allowing it to cool down to room temperature. An SU-8 3010 (MicroChem Inc) thin film was then spin-coated onto the wafer to a height of $10\,\mu m$ (spin coating parameters: 15 s at 500 rpm, followed by 30 s at 3000 rpm), followed by a soft-bake (1 min at 65°C, followed by 3 min at 95°C, and 1 min at 65°C), exposed for 5 s to UV light through the chrome mask with a mask aligner (Kloé UV-KUB 3) according to the manufacturer recommendations, and post-exposure bake (1 min at 65°C, followed by 3 min at 95°C , and 1 min at 65°C). The wafers were then developed with mr-Dev 600 (microresist technologies). Prior to their use, the master molds were treated with 1H,1H,2H,2H-perfluorodecyltriethoxysilane 97% (abcr) to reduce PDMS adhesion upon usage.

### Microsystem fabrication
The microfluidic device was produced with polydimethylsiloxane (PDMS) via soft lithography. Briefly, PDMS elastomer monomer and curing agent (Sylgard 184, Dow Corning) were mixed in a ratio 10:1 and then degassed. The PDMS was casted onto the master mold to a height of 5 mm and was cured at 80 for 2 hr. The inlets for the fluidic channel were then punched with a 1.5 mm diameter biopsy punch (pmfmedical). The PDMS was bonded to a clean glass slide by plasma activation to finish the microfluidic system.

## Swimming in sediment-like obstacle channels
### Bacterial culture
*Magnetospirillum gryphiswaldense* MSR-1 (*Schleifer et al., 1991*) was cultured in MSR-1 growth medium with the composition indicated by *Heyen and Schüler, 2003*, with the addition of pyruvate (27 mM) as carbon source instead of lactate. For the creation of an aerotactic band and the subsequent selection of swimming bacteria, the growth medium was supplemented with 0.1% agar. Briefly, 1 mL of bacteria was inoculated into the bottom of a 15 mL Hungate tube filled with 10 mL of MSR-1 growth medium with 0.1% agar. The tube was sealed with a rubber cap pierced with a needle capped with a 0.2 µm filter (Whatman) to allow the formation of an oxygen gradient. To select the North-seeking bacteria (i.e. swimming with the North of their magnet at their front in oxic conditions) in the formed band, the tube was put inside a pair of coils to apply a magnetic field parallel to the oxygen gradient, but pointing downwards to the anoxic region (equivalent to the situation that the bacteria experience in their natural habitat in the Northern Hemisphere). The bacteria were grown at 28°C and allowed to form an aerotactic band. Once the band had formed, the motile bacteria were selected by aspirating the band with a needle and culturing them for two passes in standard MSR-1 growth medium and microaerobic conditions. Subsequently, the magnetic bacteria were collected with a needle by placing a magnet next to the tube to attract them. The final population of North-seeking bacteria was estimated to be about 80% of the population. In oxic conditions such as the ones in our experiment (where oxygen can freely diffuse through the PDMS into the microchannel), these bacteria swim with the North of their magnet at their front. The optical density was measured at 565 nm ($OD_{565}$) and was adjusted as needed with fresh growth medium for subsequent measurements.

## Operation of the chip

The microfluidic system was filled with a syringe pump with MSR-1 standard growth medium at a flow rate of 15 µL·min⁻¹ and was incubated at room temperature for 30 min prior to its use. Subsequently, 25 µmL of MSR-1 standard growth medium containing bacteria at an $OD_{565}$ of 0.1 were placed in the inlet and the bacteria were allowed to swim into the microchannel, placed under a custom-designed magnetic microscope equipped with three pairs of Helmholtz coils (**Bennet et al., 2014**). Different magnetic field intensities ($B = 0, 50, 500$ µT) were applied parallel to the microchannel length. Bacteria were imaged with a 10 x objective using phase contrast microscopy.

## Monitoring of bacterial throughput

We counted the number of bacteria in the entry and exit regions of the channels (**Figure 5A**) in snapshots taken every 20 s in alternating order and summed the number in the exit region over all snapshots up to 30 min to obtain an estimate of the cumulative arrivals. These were normalized to the likewise cumulated counts in the entry region to account for differences in bacterial loading, resulting in the OUT:IN ratio plotted as a measure of throughput. The cumulated count was used in the normalization, because bacteria continue to enter into the entry region from the inlet during the experiment (this is different in our simulations, where all bacteria start at time zero at the beginning of the channel). The distribution of IN:OUT ratios at different field strengths were compared with a Mann-Whitney U test, pooling the results for different channels and excluding outliers with z-score> 2.

## Model and simulations

### Equations of Motion

We model a magnetotactic bacterium as an active Brownian sphere in two dimensions that has a permanent dipole moment $\boldsymbol{\mu} = \mu\hat{\boldsymbol{e}}$ with the magnetic strength $\mu$ at the particle center (**Telezki and Klumpp, 2020**; **Codutti et al., 2019**). The magnetic moment is aligned with the orientation of the particle $\hat{\boldsymbol{e}}$, which defines the direction of self-propulsion with speed $v_0$. The orientation of the particle is described by an angle $\varphi$ in the 2d plane. The orientation vector is then given by $\hat{\boldsymbol{e}} = (\cos\varphi, \sin\varphi)$ and the position vector is given by $\boldsymbol{r} = (x, y)$. The two-dimensional equations of motion are then

$$\dot{\boldsymbol{r}} = v_0\hat{\boldsymbol{e}} + \frac{1}{\gamma_T}\boldsymbol{F} + \sqrt{2D_T}\boldsymbol{\xi}^T \tag{1}$$

$$\dot{\varphi} = \frac{1}{\gamma_R}\tau + \sqrt{2D_R}\xi^R, \tag{2}$$

where $\gamma_T$ and $\gamma_R$ are the translational and rotational drag coefficients, $\boldsymbol{\xi}^T$ is the translational stochastic force, and $\xi^R$ is the rotational stochastic torque. Both are described by Gaussian white noise with zero mean

$$\left\langle \boldsymbol{\xi}^{T,R} \right\rangle = 0$$

$$\left\langle \boldsymbol{\xi}^{T,R}(t) \cdot \boldsymbol{\xi}^{T,R}(t') \right\rangle = \mathbf{1}\delta(t - t').$$

$D_T$ and $D_R$ are the corresponding diffusion coefficients.

$\boldsymbol{F}$ and $\tau$ describe forces and torques acting on the particle. In this study, we do not include particle-particle interactions. Forces $\boldsymbol{F}$ thus result solely from interactions with the obstacles representing the sand grains and with the confining walls of the channel. These forces are described by a Weeks-Chandler-Andersen potential between the particle and a virtual particle inside the obstacle (i.e. sand obstacle or wall). The virtual particle is located with its surface at shortest distance to the particle (i.e. with its center at a distance $\frac{\sigma}{2}$ from the surface). $\boldsymbol{r}_S$ connects the virtual particle to the particle. The repulsive surface force $\boldsymbol{F}^S$ is then derived from the potential:

$$\boldsymbol{F}^S = \begin{cases} 48\epsilon\frac{\vec{r}_S}{r_S^2}\left[\left(\frac{\sigma}{r_S}\right)^{12} - \frac{1}{2}\left(\frac{\sigma}{r_S}\right)^6\right] & \text{if } r_S < 2^{1/6}\sigma \\ 0 & \text{if } r_S \geq 2^{1/6}\sigma. \end{cases} \tag{3}$$

**Table 1.** Model parameters and their values.

| Parameter | Symbol | Value |
|---|---|---|
| Diameter of active particle | $\sigma$ | 1 μm (***Klumpp and Faivre, 2016***) |
| Magnetic moment | $\mu$ | $4 \times 10^{-16}$ J/T (***Codutti et al., 2022***; ***Nadkarni et al., 2013***) |
| Self-propulsion speed | $v_0$ | 50 μm/s |
| Fluid temperature | $T$ | 298 K |
| Fluid viscosity | $\eta$ | $8.9 \times 10^{-4}$ Pa/s |
| Magnetic field strength | $B$ | [0 . . . 500] μT |
| Surface force parameter | $\epsilon$ | 4 $k_B T$ |
| Surface torque parameter | $\alpha$ | 0.2 μm |

Here, $\epsilon$ calibrates the strength of the force.

The torques $\tau$ acting on the particle result from interactions with the obstacles ($\tau_S$) and the applied external magnetic field ($\tau_B$). The external magnetic field has magnitude $B$ and is applied in the direction of the exit of the channel. Because the field is homogeneous, particles experience no force but an aligning torque $\tau_B = \mu B \sin\varphi$. In addition, a torque arises from steric and hydrodynamic interactions with the surfaces of the obstacles and channel walls (***Volpe et al., 2011***; ***Kantsler et al., 2013***; ***Bechinger et al., 2016***; ***Ostapenko et al., 2018***). To model the reorientation of the bacteria near a surface, we introduce a surface torque,

$$\tau^S = \alpha \left[ \hat{e} \times \boldsymbol{F}^S \right]_z. \tag{4}$$

with a tuning parameter $\alpha$ (with dimension of length), which we determine by matching the sliding distances along an obstacle wall to experimental observations.

For a quantitative comparison between numerical simulations and experimental data, we used the parameter values listed in ***Table 1***.

## Simulations

We solve the equations of motion in two dimensions by using overdamped Brownian dynamics (BD) simulations. The following reduced units were used: time $t^* = t D_0^T/\sigma^2$, where $D_0^T = k_B T/3\pi\eta\sigma$ is the translational diffusion constant; position $\boldsymbol{r}^* = \boldsymbol{r}/\sigma$; $\epsilon^* = \epsilon/k_B T$; field strength $B^* = B/B_0$, where $B_0 = 1 \times 10^{-5}$T is the order of magnitude of the magnetic field strength of the earth; dipole strength $\mu^* = \mu B_0/k_B T$. The dimensionless equations of motion were integrated for $N_{\text{steps}} = 5 \times 10^6$ time steps using the Euler-Maruyama method (***Kloeden and Platen, 1992***) with a time step $\Delta t^* = 2 \times 10^{-5}$. We note that, with this choice of dimensionless units, the self-propulsion velocity $v_0^* = v_0\sigma/D_0^T$ is identical to the Péclet number.

For simulations of swimming through obstacle channels, we use spherical obstacles with the same positions and diameters as the cylindrical pillars in our microfluidic channels. We simulated all 10 different obstacle arrays. In each case, we initiated 2000 bacteria at position $x = 0$ corresponding to the IN region and tracked their motion until they arrived at the end of the channel (with $x \geq 3495\,\mu\text{m}$) or until the maximal simulated time (30 min).

For simulations of the escape from traps, traps with random parameters were constructed from two spherical obstacles as shown in ***Figure 7a and d***. We simulated the escape of 100 bacteria from each trap to determine the escape time.

## Sliding measurements and parameterization of obstacle interactions in the simulations

***Figure 4b*** illustrates how we determined the sliding distance. The sliding distance is defined as the arc length (solid green arrow) that is covered by the center of mass trajectory (solid blue line) inside the sliding region (dotted green line). The sliding region was chosen to be slightly larger than the interaction region (orange dotted line), where $F^S > 0$ and thus $\tau^S > 0$. The sliding region was set to

$2\,\mu m$, when measured from the obstacles' surface. This choice allows to compare data from numerical simulations to the experiments. It corresponds to the spatial resolution $\Delta x \approx 2\,\mu m$ of the microscope, which limits the precision with which the beginning and the end of contact with an obstacle surface can be identified.

Following that protocol, we measured the sliding distances in the absence of a magnetic field ($B = 0$) in the numerical simulations and experiment (*Figure 4c*) For $\alpha = 0.2\,\mu m$, the simulations match the experimentally measured sliding distances (*Figure 4*).

## Acknowledgements

Funding from the Deutsche Forschungsgemeinschaft (DFG, German Research Foundation – project ID 446142122) and the ANR (grant PRCI ANR-20-CE92-0051 - project Manaconv) is acknowledged. A C and E CD. were supported by the IMPRS on Multiscale Biosystems. The simulations were run on the GoeGrid cluster at the University of Göttingen, which is supported by the DFG (project IDs 436382789; 493420525) and MWK Niedersachsen (grant no. 45-10-19-F-02). We acknowledge support by the Open Access Publication Funds/transformative agreements of the University of Göttingen.

## Additional information

### Funding

| Funder | Grant reference number | Author |
| --- | --- | --- |
| Deutsche Forschungsgemeinschaft | 446142122 | Stefan Klumpp |
| Agence Nationale de la Recherche | ANR-20-CE92-0051 | Damien Faivre |
| IMPRS on Multiscale Biosystems | | Agnese Codutti Elisa Cerdá-Doñate |
| Deutsche Forschungsgemeinschaft | 436382789 | Stefan Klumpp |
| MWK Niedersachsen | 45-10-19-F-02 | Stefan Klumpp |

The funders had no role in study design, data collection and interpretation, or the decision to submit the work for publication.

### Author contributions

Agnese Codutti, Conceptualization, Software, Investigation, Writing – review and editing; Mohammad A Charsooghi, Conceptualization, Investigation, Visualization, Writing – original draft, Writing – review and editing; Konrad Marx, Investigation, Visualization, Writing – original draft, Writing – review and editing; Elisa Cerdá-Doñate, Conceptualization, Resources, Investigation, Methodology, Writing – review and editing; Omar Muñoz, Software, Investigation; Paul Zaslansky, Resources, Investigation, Writing – review and editing; Vitali Telezki, Software, Investigation, Writing – original draft, Writing – review and editing; Tom Robinson, Conceptualization, Resources, Supervision, Methodology, Writing – review and editing; Damien Faivre, Conceptualization, Supervision, Funding acquisition, Writing – review and editing; Stefan Klumpp, Conceptualization, Supervision, Funding acquisition, Writing – original draft, Writing – review and editing

### Author ORCIDs

Tom Robinson (ID) https://orcid.org/0000-0001-5236-7179
Damien Faivre (ID) https://orcid.org/0000-0001-6191-3389
Stefan Klumpp (ID) https://orcid.org/0000-0003-0584-2146

Reviewer #1 (Public review): https://doi.org/10.7554/eLife.98001.3.sa1
Reviewer #2 (Public review): https://doi.org/10.7554/eLife.98001.3.sa2
Author response https://doi.org/10.7554/eLife.98001.3.sa3

## Additional files

### Supplementary files
MDAR checklist

### Data availability
Data and code is available through Edmond at https://doi.org/10.17617/3.KWU1HZ.

The following dataset was generated:

| Author(s) | Year | Dataset title | Dataset URL | Database and Identifier |
|---|---|---|---|---|
| Codutti A, Charsooghi M, Marx K, Cerdá Doñate E, Munoz O, Zaslansky P, Teletzki V, Robinson T, Faivre D, Klumpp S | 2024 | Escape problem of magnetotactic bacteria - Physiological magnetic field strength help magnetotactic bacteria navigate in simulated sediments (Version 1) | https://doi.org/10.17617/3.KWU1HZ | Edmond, 10.17617/3.KWU1HZ |

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
