## [Editor Report · eLife Assessment]

This study presents **valuable** experimental and numerical results on the motility of a magnetotactic bacterium living in sedimentary environments, particularly in environments of varying magnetic field strengths. The evidence supporting the claims of the authors is **compelling** and the study will be of specific relevance to biophysicists interested in bacterial motility.

---

## [Referee Report · Reviewer #1 (Public review)]

Summary:

The authors present experimental and numerical results on the motility *Magnetospirillum gryphiswaldense* MSR-1, a magnetotactic bacterium living in sedimentary environments. The authors manufactured microfluidic chips containing three-dimensional obstacles of irregular shape, that match the statistical features of the grains observed in the sediment via micro-computer tomography. The bacteria are furthermore subject to an external magnetic field, whose intensity can be varied. The key quantity measured in the experiments is the throughput ratio, defined as the ratio between the number of bacteria that reach the end of the microfluidic channel and the number of bacteria entering it. The main result is that the throughput ratio is non-monotonic and exhibits a maximum at magnetic field strength comparable with Earth's magnetic field. The authors rationalize the throughput suppression at large magnetic fields by quantifying the number of bacteria trapped in corners between grains.

Strengths:

While magnetotactic bacteria general motility in bulk has been characterized, we know much less about their dynamics in a realistic setting, such as a disordered porous material. The micro-computer tomography of sediments and their artificial reconstruction in a microfluidic channel is a powerful method that establishes the rigorous methodology of this work. This technique can give access to further characterization of the microbial motility. The coupling of experiments and computer simulations lends considerable strength to the claims of the authors, because the model parameters (with one exception) are directly measured in the experiments.

Weaknesses:

The main weakness of the manuscript pertains to the comparison between simulations and experiments due to limitations in the tracking of bacteria in the experiments.

Impact:

Building on the present work, and refining the experimental setup may shed light on the microbial interactions in an environment such as soil which deserves further studies.

---

## [Referee Report · Reviewer #2 (Public review)]

Summary:

The manuscript reports results of a combined experimental and simulation study of magnetotactic bacteria in microfluidic channels containing sediment-mimicking obstacles. The obstacles were produced based on micro-computer tomography reconstructions of bacteria-rich sediment samples. The swimming of bacteria through these channels is found experimentally to display the highest throughput for physiological magnetic fields. Computer simulations of active Brownian particles, parameterized based on experimental trajectories are used to quantify the swimming throughput in detail. Similar behavior as in experiments is obtained, but also considerable variability between different channel geometries. Swimming at strong field is impeded by the trapping of bacteria in corners, while at weak fields the direction of motion is almost random. The trapping effect is confirmed in the experiments, as well as the escape of bacteria with reducing field strength.

Strengths:

This is a very careful and detailed study, which draws its main strength from the fruitful combination of construction of novel microfluidic devives, their use in motility experiments, and simulations of active Brownian particles adapted to the experiment.

Based on their results, the authors hypothesize that magnetotactic bacteria may have evolved to produce magnetic properties that are adapted to the geomagnetic field in order to balance movement and orientation in such crowded environments. They provide strong arguments in favor of such a hypothesis.

Weaknesses:

Some of the issues touched upon here have been studied also in other articles. It would be good to extend the list of references accordingly and discuss the relation briefly in the text.

Comments on revisions:

In their rebuttal letter, the authors have responded in detail to all points raised in my previous report. They have revised their manuscript accordingly.

---

## [Author Response]

The following is the authors’ response to the original reviews.

**eLife Assessment**
This study presents valuable experimental and numerical results on the motility of a magnetotactic bacterium living in sedimentary environments, particularly in environments of varying magnetic field strengths. The evidence supporting the claims of the authors is solid, although the statistical significance comparing experiments with the numerical work is weak. The study will be of interest to biophysicists interested in bacterial motility.

We thank the reviewers and editors for their careful reading and the constructive comments. With respect to the statement about weak statistical significance, we think that this statement mixes two separate issues, the significance of the difference between experiments at 0 and 50µT and the comparison of experiments with simulations. We have amended our manuscript to address both points as described below. The difference between the experiments at 0 and 50µT is indeed significant, and the discrepancy between experiments and simulations can be explained by unavoidable differences in the way we quantify bacterial throughput.

**Public Reviews:**

**Reviewer #1 (Public Review):**
Summary:The authors present experimental and numerical results on the motility Magnetospirillum gryphiswaldense MSR-1, a magnetotactic bacterium living in sedimentary environments. The authors manufactured microfluidic chips containing three-dimensional obstacles of irregular shape, that match the statistical features of the grains observed in the sediment via microcomputer tomography. The bacteria are furthermore subject to an external magnetic field, whose intensity can be varied. The key quantity measured in the experiments is the throughput ratio, defined as the ratio between the number of bacteria that reach the end of the microfluidic channel and the number of bacteria entering it. The main result is that the throughput ratio is non-monotonic and exhibits a maximum at magnetic field strength comparable with Earth's magnetic field. The authors rationalize the throughput suppression at large magnetic fields by quantifying the number of bacteria trapped in corners between grains.Strengths:While magnetotactic bacteria's general motility in bulk has been characterized, we know much less about their dynamics in a realistic setting, such as a disordered porous material. The micro-computer tomography of sediments and their artificial reconstruction in a microfluidic channel is a powerful method that establishes the rigorous methodology of this work. This technique can give access to further characterization of microbial motility. The coupling of experiments and computer simulations lends considerable strength to the claims of the authors, because the model parameters (with one exception) are directly measured in the experiments.Weaknesses:The main weakness of the manuscript pertains to the discussion of the statistical significance of the experimental throughput ratio. Especially when comparing results at zero and 50 micro Tesla. The simulations seem to predict a stronger effect than seen in the experiments. The authors do not address this discrepancy.

We thank the reviewer for their positive assessment and the detailed constructive remarks.

The increase in bacterial throughput between 0 and 50 µT is indeed more pronounced in the simulations than in the experiments, partly due to the fact that there is considerably more variability in the experimental data. We did two things to address this issue: (1) We performed additional statistical test addressing the difference between the experimental results at 0 and 50 µT. Indeed, the difference is only weakly significant (in contrast to the difference of either to 500µT). The increase is however consistent with the observation in the absence of obstacles in the channel, where we see a monotonous increase from 0 to 500 µT (Supp. Figure S5). We have added the test results in the caption of Fig. 3. (2) To address the difference between simulations and experiments, we added a section in Methods on how we determine the throughput and a short discussion in the Results section. The key points are that the initial condition is different in simulations and experiments and that the throughput is therefore quantified differently. This difference is due to experimental limitations: we cannot track bacteria through the whole channel and we wanted to avoid pushing them into the channel with fluid flow to avoid effects of flow on the results. As a consequence, bacteria continue to enter the IN region of the channel from the inlet during the experiment, while in the simulation, they all start at the beginning of the channel simultaneously. We expect this to mostly affect the case with diffusive transport (B=0).

**Reviewer #2 (Public Review):**
Summary:simulation study of magnetotactic bacteria in microfluidic channels containing sediment-mimicking obstacles. The obstacles were produced based on micro-computer tomography reconstructions of bacteria-rich sediment samples. The swimming of bacteria through these channels is found experimentally to display the highest throughput for physiological magnetic fields. Computer simulations of active Brownian particles, parameterized based on experimental trajectories are used to quantify the swimming throughput in detail. Similar behavior as in experiments is obtained, but also considerable variability between different channel geometries. Swimming at strong field is impeded by the trapping of bacteria in corners, while at weak fields the direction of motion is almost random. The trapping effect is confirmed in the experiments, as well as the escape of bacteria with reducing field strength.Strengths:This is a very careful and detailed study, which draws its main strength from the fruitful combination of the construction of novel microfluidic devices, their use in motility experiments, and simulations of active Brownian particles adapted to the experiment. Based on their results, the authors hypothesize that magnetotactic bacteria may have evolved to produce magnetic properties that are adapted to the geomagnetic field in order to balance movement and orientation in such crowded environments. They provide strong arguments in favor of such a hypothesis.Weaknesses:Some of the issues touched upon here have been studied also in other articles. It would be good to extend the list of references accordingly and discuss the relation briefly in the text.

We thank the reviewer for the constructive comments. We answer to the point concerning previous literature in the response to the recommendations below.

**Recommendations for the authors:**

**Reviewer #1 (Recommendations For The Authors):**
Here follows a list of points the authors should address.(1) Are additional experiments feasible to decrease the statistical noise present in Fig. 3c? At the very least, the authors should discuss the statistical significance of the results at 50 muT vis-a-vis 0 T.

See our response to Strengths/Weaknesses above

(2) The experimental setup is not immediately clear. I think that adding a panel from Fig. S1 (or a sketch thereof) would help clarify, especially in relation to the entry zone and end zone.

We are not sure what you mean. Fig. 3A already contains exactly such a panel. We have however added another supplementary figure that shows an additional detailed view of the setup (Fig. S3). In addition, we revised several figures: We have replaced Fig. S1 with a better version and exchanged the schematic view of the obstacle channel in Fig 1, removing the additional inlets that were not used in this study (also in Fig 3A), Instead we added a comment in Methods explaining their presence. Hopefully this makes the setup clear.

(3) It should be also stated that there is no external flow imposed on the channel.

We have added such a statement in the description of the experiment in section 2.2 Swimming of magnetotactic bacteria through sediment-mimicking obstacle channels.

(4) Fig. 3c and Fig. 6c are seemingly showing the same quantity (or closely related ones). The authors should use the same symbol and give an explicit mathematical definition.

The two quantities are not exactly the same, as we cannot directly quantify the flux of bacteria through the channel in our experiments. On the one hand, we cannot track bacteria through the whole channel, on the other hand, the initial conditions are not exactly the same as in the simulations. In the simulations all bacteria start at the same time at the entrance to the channel. In the experiments, they enter from the inlet and do so at different times (pushing them in with fluid flow would be possible, but carries the risk of perturbing the results due to induced flow through the channel). We have added a new section in the Methods section that explains this difference and describes the procedure used to obtain the throughput from the experiments in detail. We have also added a corresponding comment in the Result section, where the simulations are compared with the experiments.

Minor issues:- Figures have different styles that should be unified. For example, the panel labels sometimes have round brackets and sometimes they don't.

See above

- Page 6, (muCT) should have the Greek letter mu

Thanks, corrected.

- Fig. 3a is not very clear; see my point 2 above.

See above

**Reviewer #2 (Recommendations For The Authors):**
I have only a few comments and questions, which the authors should address:(1) The observed exponential dependence of decay time on the "well" depth could be related to the exponential density distribution of active particles in a gravitational field, which has been derived previously. Might be interesting to discuss such a possible connection.

Thank you for the suggestion, the two cases are indeed somewhat analogous with behaviors reminiscent of thermal processes with an effective temperature. Such a description is however not generally possible (even for sedimentation, only some features are described). We plan to address in future work whether it can be made more quantitative in our case of escape from the corner traps. We have included a short discussion of the analogy in the section on trapping and escape.

(2) The authors should consider the following relevant references, and discuss them briefly in their manuscript:- Sedimentation, trapping, and rectification of dilute bacteria J Tailleur, ME Cates EPL 86, 60002 (2009)- Human spermatozoa migration in microchannels reveals boundary-following navigation P Denissenko, V Kantsler, DJ Smith, J Kirkman-Brown Proc. Natl. Acad. Sci. USA 109, 8007-8010 (2012)- Wall accumulation of self-propelled spheres J Elgeti, G Gompper Europhysics Letters 101, 48003 (2013)- Wall entrapment of peritrichous bacteria: a mesoscale hydrodynamics simulation study SM Mousavi, G Gompper, RG Winkler Son Maber 16 (20), 4866-4875 (2020)- A Geometric Criterion for the Optimal Spreading of Active Polymers in Porous Media C Kurzthaler, S Mandal, T Bhabacharjee, H Löwen, SS Daba, HA Stone Nat. Commun. 12, 7088 (2021)- Run-to-Tumble Variability Controls the Surface Residence Times of *E. coli* Bacteria G Junot, T Darnige, A Lindner, VA Martinez, J Arlt, A Dawson, WCK Poon, H Auradou, E Clement Phys. Rev. Leb. 128, 248101 (2022)- Dynamics and phase separation of active Brownian particles on curved surfaces and in porous media P Iyer, RG Winkler, DA Fedosov, G Gompper Phys. Rev. Research 5, 033054 (2023)

We agree that there is a lot of literature on these aspects, specifically interaction of self-propelled objects with walls and motion of swimmers through porous media. We have slightly extended our overview of previous literature in the introduction and included most of these references.